# Association between Age-Friendliness of Communities and Frailty among Older Adults: A Multilevel Analysis

**DOI:** 10.3390/ijerph19127528

**Published:** 2022-06-20

**Authors:** Jixiang Xu, Yingwei Chen, Yujie Wang, Junling Gao, Limei Huang

**Affiliations:** 1School of Public Health, Fudan University, Shanghai 200032, China; 20211020057@fudan.edu.cn (J.X.); 19211020088@fudan.edu.cn (Y.C.); 21211020060@m.fudan.edu.cn (Y.W.); 2Collaborative Innovation Cooperative Unit, National Clinical Research Center for Geriatric Diseases, Shanghai 200032, China; 3Core Unit of Shanghai Clinical Research Center for Geriatric Diseases, Shanghai 200032, China; 4Songjiang Center of Disease Prevention and Control, Shanghai 201620, China; hlm950526@163.com

**Keywords:** age-friendly community, frailty, older adults, multilevel analysis

## Abstract

An age-friendly environment is one of the measures of healthy aging. However, there is scarce evidence of the relationship between the age-friendliness of communities (AFC) and frailty status among Chinese older adults. This study aims to examine this relationship using a multilevel analysis with the data of a cross-sectional study conducted among 10,958 older adults living in 43 communities in four cities in China. The validated Age-friendly Community Evaluation Scale and Chinese frailty screening-10 Scale (CFS-10) were used to measure AFC and Frailty. Multilevel regression analyses were performed to examine the relationship between the AFC in two assessments of individual- and community-level and frailty status. After controlling for individual-level socio-demographic, health status, and lifestyle variables, compared with older adults in the lowest quartile of the individual-level perception of AFC, the frailty odds ratios for those in the top three quartiles were 0.69 (95% confidence interval [CI]: 0.56–0.83), 0.75 (95% CI: 0.61–0.91), and 0.56 (95% CI: 0.48–0.74). However, there was no association between the community-level AFC and frailty. A higher level of age-friendliness in the community is associated with lower frailty odds. Therefore, building age-friendly communities may be an important measure to prevent frailty among Chinese older adults.

## 1. Introduction

The United Nations reported that the population aged over 65 years will exceed 1.5 billion and account for 16% of the total population by 2050 [1]. The world’s rapidly aging population poses substantial challenges to health and social care systems. To address these challenges, the World Health Organization (WHO) in 2015 proposed a public health framework for healthy aging [2], which defined healthy aging as “the process of developing and maintaining the functional ability that enables well-being in older age.” In contrast with the traditional perspective of aging, healthy aging pays more attention to an individual’s functional ability throughout the course of their life, which is determined by an individual’s intrinsic capacity, their environment, and the interaction of their intrinsic capacity and environment [3].

Age-friendly environments foster healthy and active aging by building and maintaining an individual’s intrinsic capacity over the course of a life and enabling greater functional ability in individuals with a given level of capacity [4]. Building age-friendly environments is a global strategy for healthy aging [5]. Frailty, a major clinical condition among the aging population, is a state of being vulnerable to multiple adverse health outcomes, including falls, institutionalization, hospitalization, fractures, disabilities, and death, due to age-associated declines in physiologic reserve and function across multiple organ systems [6,7,8,9]. Frailty is the declining state of intrinsic capacity before disability occurs during life [10]. Previous studies found that frailty is associated with biological and psychological factors, including age, gender, nutritional status, chronic diseases, depression, and cognitive function [11,12,13,14,15,16], while less attention has been paid to the effects of contextual factors. Older adults consistently prefer aging in place [17]; hence, their communities, as the main sites in which older adults live and move about, play a key role in their lives [18,19]. Studies have also shown that age-friendly communities are positively associated with life satisfaction [20], good self-rated health and a lower likelihood of functional limitations [21,22,23], and mental health [24,25]. Although previous studies have demonstrated that some community features, including walkability, green space, and social capital, are negatively related to frailty [26,27], to our knowledge, there is no research on the relationship between holistic assessment including individual- and community-level of age-friendliness of communities (AFC) and frailty. To fill this gap, the current study examined the relationship between holistic AFC and frailty in a large sample of older Chinese people.

## 2. Materials and Methods

### 2.1. Study Design and Participants

This cross-sectional study was conducted in four cities of Shanghai (Eastern China), Zhuhai (Southern China), Panzhihua (Western China), and Ordos (Northern China) from June 2020 to July 2021. Older adults aged 65 years or older, from 43 communities were randomly selected by multi-stage stratified sampling. First, 16 communities in Shanghai, 6 in Zhuhai, 12 in Panzhihua, and 9 in Ordos were conveniently selected, covering both urban and rural areas in each city. Next, at least 200 older adults were randomly selected from each selected community. Participants were eligible if they were aged 65 years or older and had lived in the local community for six months before being investigated, and had no progressive tumors or severe mental disorders. Trained interviewers from each selected community visited the participants in their homes or invited them to community healthcare centers to collect data using a face-to-face survey using a self-administered questionnaire. After collection, the data of 390 participants (182 males missing the information of marital status, education level, self-rated health status, smoking, drinking, and several items of Age-friendly Community Evaluation Scale; 208 females missing the information of marital status, self-rated health status, vegetable intake, fruit intake, physical activity, and several items of Chinese Frailty Screening-10 Scale) were excluded due to incomplete questionnaires, and data from a total of 10,958 older adults were included for the final analysis in this survey. The Ethics Committee for Medical Research at the School of Public Health, Fudan University, approved the study (IRB00002408 and FWA00002399).

### 2.2. Measurements

#### 2.2.1. Age-Friendliness of the Community

The age-friendliness of communities (AFC) was assessed by the Age-friendly Community Evaluation Scale, designed based on the person-environment fit paradigm and demonstration of good reliability and validity [28,29]. The scale includes “Housing” (3 items), “Transportation” (7 items), “Built Environment” (6 items), “Social Participation” (6 items), and “Social Inclusion and Health Services” (10 items), with each item scored on a 5-point Likert scale (1 = completely disagree, 2 = disagree, 3 = neutral, 4 = agree, and 5 = completely agree). Age-friendliness of communities was measured first by aggregating the total scores of the five dimensions and then considering the bias from the discrepancy in personal preference and utilization among individual older adults. The age-friendliness of communities was assessed in two alternative ways: (a) individual-level AFC, a calculation of the mean score of each individual’s own assessments on the corresponding scale’s items and (b) community-level AFC, an estimation of the mean scale score of all respondents in the same community. Ultimately, both the individual-level and community-level AFC were divided into quartiles for analysis, and the highest quartile indicated the highest level of the corresponding age-friendliness of communities. Specific items of the Age-friendly Community Evaluation Scale can be seen in Appendix A.

#### 2.2.2. Frailty

Frailty status was assessed using the validated Chinese Frailty Screening-10 Scale (CFS-10) [30]. The CFS-10 consists of 10 yes (1)/no (0) items covering five domains of intrinsic capacity: locomotor capacity, sensory capacity, vitality, cognition, and psychological capacity [3]. Frailty was judged present if the number of positive items reached 5 or more. A previous study [31] indicated that the CFS-10 had excellent criteria validity with the area under the receiver operating characteristic curves (AUCs) based on the Frailty Classification Scale (FRAIL), The Tilburg Frailty Index (TFI), and the Frailty Index (FI) of 0.91, 0.87, and 0.87, respectively. The diagnostic validity for disability of Instrumental Activities of Daily Living Scale (IADL) was slightly more powerful than the other three frailty instruments; AUC for the IADL disability compared with the FRAIL, TFI, and FI was 0.79 vs. 0.74, 0.75, and 0.79, respectively. Specific items of the CFS-10 can be seen in Appendix A.

#### 2.2.3. Covariates

The following variables were selected as potential confounders: age, gender, marital status, education level, self-rated health status, smoking, drinking, vegetable intake, fruit intake, physical activity, Non-communicable Disease (NCD), depression, and cognitive function. Whether the intake of vegetables and fruit was sufficient and physical activity was up to standard was based on the standards of the dietary guidelines for Chinese residents and guidelines for physical activity of Chinese people, respectively [32,33]. NCD refers to a general term for a class of diseases with concealed onset, long course of disease, prolonged illness, lack of clear evidence of infectious biological etiology, complex etiology, or incompletely confirmed etiology. In our study, it involved hypertension, diabetes, stroke, cerebral infarction, cerebral hemorrhage), ischemic heart disease, chronic lung disease, tumor or cancer (except for mild skin cancer), congestive heart failure, angina, asthma, arthritis, kidney disease, cirrhosis, chronic liver disease, gastrointestinal diseases, Parkinson, and musculoskeletal disorders. In addition, cognitive function was measured using cognitive self-assessment tools (AD8) [34] and the depressive module of the Patient Health Questionnaire (PHQ-9) [35] was applied to detect the depressive symptoms to evaluate depression of the older adults. The specific classification of each variable can be seen in Appendix A.

### 2.3. Statistical Analysis

#### 2.3.1. Descriptive analysis

Descriptive statistics with frequencies and percentages were reported. The Cochran–Mantel–Haenszel (CMH) *χ*^2^ test was applied to compare the characteristics of participants with different frailty statuses among the categorical variables to independently test the factors affecting the frailty of the older adults [36].

#### 2.3.2. Multilevel Regression Analysis

The data were from a multilevel structure comprised of older adults (at level 1) nested within communities (at level 2) in the current study. Therefore, multilevel logistic regression models were used to examine the associations between individual-level and community-level AFC and frailty and obtain the adjusted odds ratios (ORs) and their 95% confidence intervals (CIs).

This study includes four models created in accordance with the typical multilevel analysis [36]:(a)Empty model: Examined community-level variance in the prevalence of frailty, excluding any explanatory variable and obtaining the intraclass correlation coefficient (ICC) used to indicate the variances of frailty among communities. A high ICC means larger variances between communities.(b)Model 1: Separately examined the impact of community-level AFC on frailty without any individual-level variable.(c)Model 2: Examined the relationship between individual-level AFC and covariates with frailty.(d)Model 3: Simultaneously examined the relationship between individual-level and community-level AFC and frailty after controlling for covariates.

Furthermore, −2 log likelihood (−2LL), Akaike information criterion (AIC), and Bayesian information criterion (BIC) were used to compare the goodness-of-fit of each model. The SAS version 9.4 (made by the company of Statistical Analysis System, Laray, NC, USA) was used for all analyses and the multilevel analyses were performed with the NLMIXED and GLIMMIX procedures [37].

## 3. Results

### 3.1. Participant Characteristics

The demographic characteristics of the 10,958 participants from 43 communities are listed in Table 1. Among the participants, the proportion of the older adults aged 65–69 years (37.82%) was the highest. Among the four age groups, the proportion of frail older individuals in the age group of 80 and above is the highest, accounting for 23.52% and the highest proportion (91.68%) of robust older adults was in the 65–69 age group. The number of females (53.46%) was slightly greater than the number of males, and 45.28% of respondents across all communities were illiterate. Furthermore, 78.77% reported they were married and 91.18% rated their health status as general or above. Approximately 8403 (76.68%) respondents did not smoke and 9415 (85.92%) did not drink. Half of the participants ate a sufficient amount of vegetables, whereas the majority had substandard fruit intake and were not physically active. In addition, 80% of the participants suffered from chronic diseases, of which nearly half were affected by two or more. A total of 10,230 (93.36%) participants were not depressed, and nearly one-third of the participants suffered from cognitive impairment. In comparing the frail and robust groups, it could be found that in the group of female, unmarried, poorly educated, poorly self-rated health, no smoking and drinking, low vegetable and fruit intake, physically inactive, two or more NCDs, depression, and cognitive decline, the proportion of frailty was higher.

In total, 13.64% of the participants were frail, and CMH *χ*^2^ tests showed that the distribution of frailty varied significantly. The prevalence of frailty increased with years, and those aged above 80 years had the highest prevalence of frailty (23.52%). Older adults who were female, unmarried, poorly educated, had self-rated poor health, non-smokers, and non-drinkers reported a higher prevalence of frailty, with rates of 15.60%, 19.86%, 17.86%, 36.40%, 14.20%, and 14.48%, respectively. The prevalence of frailty was negatively related to both vegetable and fruit intake (both *p* < 0.001). Compared with the physically inactive participants, the prevalence of frailty among the older adults who regularly engaged in physical activities was lower (9.77%). The respondents without chronic diseases, depression, and cognitive decline exhibited the lowest prevalence of frailty, that is, 5.15%, 10.93%, and 4.33%, respectively. Regarding the age-friendliness of the community, the prevalence of frailty ranked lowest among older adults living in the community with the highest individual-level AFC (9.80%) and highest community-level AFC (12.66%).

### 3.2. Multilevel Analyses of the Associations between Age-Friendliness of the Community and Frailty

Table 2 shows the results of multilevel regression analyses of the four models. The ICC obtained by running the empty model was 0.175. This means the community-level variance accounted for 17.5% of all variation in the prevalence of frailty among older adults. Therefore, further analysis of the multilevel models was needed to accurately estimate the specific associations. First, community-level AFC was entered in the model, and no notable association with frailty was found (model 1). After controlling for all covariates, compared with older adults with the lowest quartile of individual-level AFC, the odds ratios of frailty for those with the second, third, and fourth quartiles of individual-level community age-friendliness were 0.69 (95% CI: 0.57–0.84), 0.75 (95% CI: 0.62–0.92), and 0.60 (95% CI: 0.49–0.74), respectively (model 2). When all covariates, individual- and community-level AFC were entered into the model simultaneously, the community-level AFC was also not associated with frailty. Furthermore, compared with older adults with the lowest quartile of the individual-level AFC, the frailty ratio for those with the second, third, and fourth quartiles of the individual-level AFC were 0.69 (95% CI: 0.56–0.83), 0.75 (95% CI: 0.61–0.91), and 0.59 (95% CI: 0.48–0.74), respectively (model 3). According to the −2LL and AIC, models 2 and 3 fit the sample better than model 1. However, models 2 and 3 were not significantly different from each other (*χ*^2^ = 1.8194, *p* = 0.611) according to the likelihood ratio test, indicating both models possessed good fitting. Given that the current study aimed to simultaneously examine the relationship between both individual-level and community-level AFC and frailty, model 3 was chosen.

## 4. Discussion

To the best of our knowledge, this is the first study to investigate the relationship between age-friendliness of communities and frailty among a large sample of older adults across 43 Chinese communities using multilevel analyses. In addition to confirming the relationship between frailty and individual characteristics, including age, gender, self-rated health status, chronic disease, depression, and cognitive function [11,12,13,14,15,16], we found that a higher AFC is associated with lower odds of frailty, and the individual-level perception of AFC played a more significant role for the older adults in contrast with the community-level AFC.

Social support derived from the social environment of the age-friendly communities might be considered a pathway that verifies the results of this study. The assistance and support provided from the dimensions of “Social Participation” and “Social Inclusion and Health Services” in the community, showed that communities promote interpersonal communication, and help community members to build a comprehensive social network via which healthy lifestyle changes, such as the spread of smoking cessation, are promoted [38,39,40,41]. In relation to this study, in communities with a higher AFC, wherein activities are rich and varied, even individuals who do not take an active part in activities may have opportunities to participate in or observe occasional events. It may help consolidate their own social support network to avoid loneliness [40]. Furthermore, the communities with a higher AFC are keen on bringing about an atmosphere that respects the social value of the older adults and encourages their social integration which has a positive impact on their health [42,43]. Thus, there may be better underlying social support in those communities with a higher individual-level AFC, which may lead to a reduced risk of frailty.

Another possible pathway to explain the association of AFC with frailty is physical activity participation, which has been investigated as a solution to slow the development of frailty [44]. Communities with a higher AFC are more likely to have better physical environments in the areas of “Housing,” “Transportation,” and “Built Environment” that are equipped with safe facilities suitable for older individuals to exercise and establish sports groups. This can result in enhancing their enthusiasm for physical activities and has positive effects on health [45,46].

Previous studies have reported that the community is the main arena for older adults to participate in daily life and activities, and whether the community enables them to safely and effectively engage with it is of vital importance [18,47,48]. Thus, this research suggests that a community’s age-friendliness, particularly the individual-level AFC, is positively associated with robustness among older adults, albeit to varying degrees. The person–environment fit paradigm may explain the underlying mechanism, which emphasizes that a person’s health and independent aging ability depend on the relationship between their internal ability and the environment [19]; this also suggests the essential importance of paying attention to the combination of the environment and the internal needs of older adults. Consequently, the multilevel analysis results of this study found that the prominent association between the individual-level (rather than the community-level) assessment of AFC and frailty, may be due to the variety of individual preferences, accessibility, and availability of the community resources used. Older adults with a higher individual-level AFC might attain better incentives and access to participation in community activities and use of community resources, which might make them have stronger health autonomy and better health status. Although the community-level AFC varied among different communities, the older adults were not active in social participation or were unable to effectively use various resources in the community anyway; not to mention generating a sense of trust and dependence, the impact of the relationship between the community-level AFC and frailty may be overlooked. Furthermore, multi-layer nesting, wherein a low-level is always simultaneously nested in multiple high-levels, may be another reason resulting in no prominent association found between the community-level AFC and frailty. Owing to the present limitation that the model of multilevel regression analysis applied in this study required that a lower level could only be nested in one higher level, whereas other high-level factors, such as families or districts might also influence frailty, the effects of community-level AFC on frailty was underestimated.

However, the study found no correlation between marital status, education level, smoking, drinking, vegetable intake, fruit intake, physical activity and frailty. It may be due to the fact that the numbers divided into the specific groups of the variables have obviously big differences and needed more accurate and reasonable grouping, which makes the group division not very reasonable for comparison and the results are not ideal. At the same time, these variables are mostly health behaviors, and their impact on frailty may need to be monitored in cohort studies rather than cross-sectional studies to carry out a better analysis. Therefore, future research can be targeted in this area.

A strength of this research is that it provides confirmation of the association between AFC and frailty in a large sample size, and it did so by investigating the data of older adults in four cities with differing economic development, which is of scientific and practical significance. Moreover, this research offers empirical evidence by quantifying the extent to which AFC at both the community and individual levels contributes to frailty, after accounting for other variables known to affect frailty. The results describe the independent and simultaneous relationship between the two levels and frailty to excavate the key points and difficulties of the development of age-friendly communities, with the aim of providing an advisory opinion on its management both in theory and practice. Key limitations to our study should also be noted. First, because of the cross-sectional design, reverse causality could potentially account to some degree for the observed associations, that is, frailty contributing to lower individual-level and community-level AFC. Second, in view of the different emphasis on the construction of age-friendly communities in different countries and regions, we cannot make the same recommendations for developing age-friendliness across all the communities; however, this study provides a supplementary reference to exploring the specific impact mechanism of different age-friendly communities on older residents’ health in China and around the world.

## 5. Conclusions

It can be concluded that older individuals who live in communities with high levels of age-friendliness are less likely to be frail than those who live in communities with lower levels of age-friendliness. Therefore, it is crucial to prevent frailty and facilitate healthy aging by building age-friendly communities with improved housing, transportation, and built environment, along with encouraging social participation and promoting the inclusion of social and health services.

## Figures and Tables

**Table 1 ijerph-19-07528-t001:** Demographic characteristics and frail status of the study subjects.

Characteristics	N (%)	Frail Status		*χ* ^2^	*p*-Value
		Robust	Frail		
**Total**	10,958 (100.00)	9463 (86.36)	1495 (13.64)		
**Age (years)**				253.69	<0.0001
65~	4085 (37.28)	3745 (91.68)	340 (8.32)		
70~	3205 (29.25)	2787 (86.96)	418 (13.04)		
75~	2150 (19.62)	1770 (82.33)	380 (17.67)		
80~	1518 (13.85)	1161 (76.48)	357 (23.52)		
**Gender**				41.02	<0.0001
Male	5100 (46.54)	4519 (88.61)	581 (11.39)		
Female	5858 (53.46)	4944 (84.40)	914 (15.60)		
**Marital status**				96.94	<0.0001
Not in marriage	2326 (21.23)	1864 (80.14)	462 (19.86)		
In marriage	8632 (78.77)	7599 (88.03)	1033 (11.97)		
**Education level**				103.57	<0.0001
Below Primary School	4962 (45.28)	4076 (82.14)	886 (17.86)		
Primary School	3419 (31.20)	3050 (89.21)	369 (10.79)		
Junior high school	1771 (16.16)	1617 (91.30)	154 (8.70)		
High school and above	806 (7.36)	720 (89.33)	86 (10.67)		
**Self-rated health**				792.54	<0.0001
Poor or Worse	967 (8.82)	615 (63.60)	352 (36.40)		
General	4194 (38.27)	3383 (80.66)	811 (19.34)		
Good	4492 (40.99)	4217 (93.88)	275 (6.12)		
Perfect	1305 (11.91)	1248 (95.63)	57 (4.37)		
**Smoking**				8.52	0.0035
No	8403 (76.68)	7210 (85.80)	1193 (14.20)		
Used to	658 (6.00)	581 (88.30)	77 (11.70)		
Smoking	1897 (17.31)	1672 (88.14)	225 (11.86)		
**Drinking**				39.46	<0.0001
No	9415 (85.92)	8052 (85.52)	1363 (14.48)		
Drinking	1543 (14.08)	1411 (91.45)	132 (8.55)		
**Vegetable intake**				186.94	<0.0001
Low	5434 (49.59)	4447 (81.84)	987 (18.16)		
Adequate	5524 (50.41)	5016 (90.80)	508 (9.20)		
**Fruit intake**				88.40	<0.0001
Low	8375 (76.43)	7089 (84.64)	1286 (15.36)		
Adequate	2583 (23.57)	2374 (91.91)	209 (8.09)		
**Physical activity**				97.45	<0.0001
Inactive	6446 (58.82)	5392 (83.65)	1054 (16.35)		
Active	4512 (41.18)	4071 (90.23)	441 (9.77)		
**Chronic disease**				394.82	<0.0001
Without NCD	2173 (19.83)	2061 (94.85)	112 (5.15)		
With one disease	3204 (29.24)	2956 (92.26)	248 (7.74)		
With Two or more	5581 (50.93)	4446 (79.66)	1135 (20.34)		
**Depression**				962.86	<0.0001
No	10,230 (93.36)	9112 (89.07)	1118 (10.93)		
Yes	728 (6.64)	351 (48.21)	377 (51.79)		
**Cognitive Function**				1668.77	<0.0001
Normal	7388 (67.42)	7068 (95.67)	320 (4.33)		
Cognitive decline	3570 (32.58)	2395 (67.09)	1175 (32.91)		
**Individual-level AFC**				124.71	<0.0001
First quartile	2812 (25.66)	2227 (79.20)	585 (20.80)		
Second quartile	2841 (25.93)	2510 (88.35)	331 (11.65)		
Third quartile	2672 (24.38)	2351 (87.99)	321 (12.01)		
Fourth quartile	2633 (24.03)	2375 (90.20)	258 (9.80)		
**Community-level AFC**				14.18	0.0002
First quartile	1957 (17.86)	1620 (82.78)	337 (17.22)		
Second quartile	3371 (30.76)	2936 (87.10)	435 (12.90)		
Third quartile	2874 (26.23)	2500 (86.99)	374 (13.01)		
Fourth quartile	2756 (25.15)	2407 (87.34)	349 (12.66)		

**Table 2 ijerph-19-07528-t002:** Multilevel results predicting frailty by individual-level and community-level AFC.

	Empty Model	Model 1		Model 2		Model 3	
		OR (95%Cl)	*p*-Value	OR (95%Cl)	*p*-Value	OR (95%Cl)	*p*-Value
Intercept				0.05 (0.03~0.08)	<0.001	0.05 (0.03~0.08)	<0.001
**Age (years)**							
65~				1.00		1.00	
70~				1.35 (1.13~1.61)	0.001	1.35 (1.13~1.61)	0.001
75~				1.56 (1.29~1.88)	<0.001	1.56 (1.29~1.88)	<0.001
80~				1.93 (1.56~2.38)	<0.001	1.93 (1.56~2.38)	<0.001
**Gender**							
Male				1.00		1.00	
Female				1.23 (1.05~1.46)	0.014	1.23 (1.04~1.45)	0.015
**Marital status**							
Not in marriage				1.00		1.00	
In marriage				1.02 (0.87~1.19)	0.833	1.02 (0.87~1.19)	0.835
**Education level**							
Below Primary School				1.00		1.00	
Primary School				1.01 (0.86~1.20)	0.89	1.01 (0.85~1.19)	0.917
Junior high school				0.95 (0.76~1.20)	0.684	0.95 (0.75~1.19)	0.651
High school and above				1.09 (0.81~1.48)	0.56	1.09 (0.80~1.48)	0.584
**Self-rated health**							
Poor or Worse				1.00		1.00	
General				0.57 (0.47~0.70)	<0.001	0.57 (0.47~0.70)	<0.001
Better				0.31 (0.25~0.39)	<0.001	0.31 (0.25~0.39)	<0.001
Perfect				0.15 (0.11~0.22)	<0.001	0.15 (0.11~0.22)	<0.001
**Smoking**							
No				1.00		1.00	
Used to				0.95 (0.70~1.30)	0.765	0.95 (0.70~1.30)	0.763
Smoking				1.19 (0.97~1.46)	0.092	1.19 (0.97~1.47)	0.092
**Drinking**							
No				1.00		1.00	
Yes				0.92 (0.72~1.17)	0.49	0.92 (0.72~1.17)	0.499
**Vegetable intake**							
Low				1.00		1.00	
Adequate				0.87 (0.74~1.02)	0.075	0.86 (0.74~1.01)	0.073
**Fruit intake**							
Low				1.00		1.00	
Adequate				0.99 (0.81~1.22)	0.926	0.99 (0.81~1.22)	0.958
**Physical activity**							
Inactive				1.00		1.00	
Active				0.91 (0.78~1.06)	0.235	0.91 (0.78~1.06)	0.224
**Chronic disease**							
Without NCD				1.00		1.00	
With one disease				1.28 (0.99~1.66)	0.06	1.28 (0.99~1.66)	0.061
With Two or more				2.35 (1.86~2.98)	<0.001	2.35 (1.86~2.97)	<0.001
**Depression**							
No				1.00		1.00	
Yes				3.54 (2.90~4.32)	<0.001	3.54 (2.90~4.32)	<0.001
**Cognitive Function**							
Normal				1.00		1.00	
Cognitive decline				6.01 (5.14~7.02)	<0.001	6.01 (5.14~7.02)	<0.001
**Individual-level AFC**							
First quartile				1.00		1.00	
Second quartile				0.69 (0.57~0.84)	<0.001	0.69 (0.56~0.83)	<0.001
Third quartile				0.75 (0.62~0.92)	0.005	0.75 (0.61~0.91)	0.005
Fourth quartile				0.6 (0.49~0.74)	<0.001	0.59 (0.48~0.74)	<0.001
**Community-level AFC**							
First quartile		1.00				1.00	
Second quartile		3.22 (1.08~10.7)	0.744			1.06 (0.67~1.69)	0.789
Third quartile		3.67 (2.13~9.62)	0.848			1.18 (0.74~1.88)	0.486
Fourth quartile		2.01 (1.43~4.01)	0.384			1.08 (0.65~1.77)	0.765
**ICC**	0.175						
**−2LL ^1^**	8145.4	8140.8		6206.6		6204.8	
**AIC ^2^**	8149.4	8150.7		6258.6		6262.8	
**BIC ^3^**	8164.0	8187.3		6448.5		6474.6	

^1^ −2LL: −2 Log Likelihood (smaller is better); ^2^ AIC: Akaike information criterion (smaller is better); ^3^ BIC: Bayesian information criterion (smaller is better).

## Data Availability

Not applicable.

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
