# Peer review of "Association between Age-Friendliness of Communities and Frailty among Older Adults: A Multilevel Analysis"

_ijerph, 2022, doi:10.3390/ijerph19127528_

Round 1

Reviewer 1 Report

This work of Xu et al was very interesting and describe the role of age-friendliness of communities among older adults.

Major 

1) Concerning the exclusion of 390 elders is it possible to add more information at least age and gender to understand if there is some motivation about their missing since the interviews were made by interviewers face to face?

2) Line 84 please give details about the 5 point likert (e.g. 1=completely disagree... to 5=...)

3) Regarding the choice to run Cochran-MantelHaenszel chi square (repeated test of indipendence) you need to provide details to explain your "statistical" reasons. Regarding the choice to run Cochran-MantelHaenszel chi square (repeated test of independence) you need to provide details to explain your reasons "statistics". Did you use it because you had to repeat the same comparisons as if it were a final sample made up of 4 different samples of the 4 selected areas?

4) The Table1 reported the % of rows but if we compare robust vs frail it will be better to have an idea inside the two groups compared. So consequently you need to review what you have reported into the manuscript.

5) You have to justify whether the responses of people with cognitive decline, who are 35% of the sample, can be reliable and properly understood together with the explanation to how prevent memory bias.

6) 172-174 lines: "The 171 ICC obtained by running the empty model was 0.175. This means the community-level variance accounted for 17.5% of all variation in the prevalence of frailty among the older adults, indicating that the frailty was related to the community they live in." 

This is a comment of the result. Please justify more clearly and with some methodology  why the frailty was related to the community with a value of 17.5% of ICC.

7) All the comments related to result should be reported in the discussion section

8) Have you think to perform your models between elders with normal cognitive status vs elders with cognitive impair?

Minor 

Please move in line 57 the extended version of ACF reported inline 79

It would also be nice  to report some charcteristics of 390 elders excluded (for example mean age and % of gender) to describe better who they are  and, possibility would also be nice to report some characteristics of about 390 elderly people (for example the average age and % of the gender) to better describe who they are and, possibly, to compare if there were differences with the final sample.

It will be better to change in line 257 the verb "demonstrate" with "describe".

Author Response

Response to reviewer 1 comments is shown in the following document.

Reviewer 2 Report

1.     Line 110: definition of chronic disease is unclear. What is included in this variable? Depression also needs more information since it is a complex disease.

2.     Page 6 and 7, NCD is not defined, and this abbreviate is not mentioned before use.

3.     Education level is not a significant predictor. However, the social-economic status seems to be associated with frailty. It is pretty counterintuitive. More explanation and discussion could help readers.

Ref: J Epidemiol Community Health. 2010 Jan; 64(1): 63–67.

doi: 10.1136/jech.2008.078428. PMCID: PMC2856660. NIHMSID: NIHMS183799. PMID: 19692719. “Socioeconomic Status is associated with Frailty: the Women’s Health and Aging Studies”

Author Response

Response to reviewer 2 comments is shown in the following document.

Round 2

Reviewer 1 Report

Thanks to the authors, the manuscript is now better than the previous version. Some statistical details are reported only in the author's reply, it will be better to enrich with at least some bibliography lines 134-135(Cochran-Mantel Haenszel).

Author Response

Response to reviewer 1 comments (round 2) is shown in the following document.
